Removing batch effects for prediction problems with frozen surrogate variable analysis

Parker Hilary S. 1
Corrada Bravo Héctor 2
Leek Jeffrey T. 1 jleek@jhsph.edu
1 Department of Biostatistics, Johns Hopkins Bloomberg School of Public Health , Baltimore, MD , USA
2 Center for Bioinformatics and Computational Biology, Department of Computer Science, University of Maryland , College Park, MD , USA
Wilke Claus
Electronic publication date: 2014 Sep 23
Publication date: 2014
Volume: 2
Electronic Location ID: e561
Received 2014 Jun 20; Accepted 2014 Aug 15
Copyright: © 2014 Parker et al.
Copyright year: 2014
Copyright holder: Parker et al.
License: This is an open access article distributed under the terms of the Creative Commons Attribution License, which permits unrestricted use, distribution, reproduction and adaptation in any medium and for any purpose provided that it is properly attributed. For attribution, the original author(s), title, publication source (PeerJ) and either DOI or URL of the article must be cited.
License URL: https://creativecommons.org/licenses/by/4.0/

Keywords: Batch effects, Surrogate variable analysis, Prediction, Machine learning, Database, Statistics, Genomics

Funding: NIH R01GM105705 R01GM103552 JTL was partially supported by R01GM105705 and R01GM103552. HIP was partially supported by a Bloomberg Fellowship. The funders had no role in study design, data collection and analysis, decision to publish, or preparation of the manuscript.

==============================
Batch effects are responsible for the failure of promising genomic prognostic signatures, major ambiguities in published genomic results, and retractions of widely-publicized findings. Batch effect corrections have been developed to remove these artifacts, but they are designed to be used in population studies. But genomic technologies are beginning to be used in clinical applications where samples are analyzed one at a time for diagnostic, prognostic, and predictive applications. There are currently no batch correction methods that have been developed specifically for prediction. In this paper, we propose an new method called frozen surrogate variable analysis (fSVA) that borrows strength from a training set for individual sample batch correction. We show that fSVA improves prediction accuracy in simulations and in public genomic studies. fSVA is available as part of the sva Bioconductor package.

Introduction

Genomic technologies were originally developed and applied for basic science research and hypothesis generation (Eisen et al., 1998). As these technologies mature, they are increasingly being used as clinical tools for diagnosis or prognosis (Chan & Ginsburg, 2011). The high-dimensional measurements made by microarrays can be used to classify patients into predictive, prognostic, or diagnostic groups. Despite the incredible clinical promise of these technologies there have only been a few signatures that have successfully been translated into the clinic.

One of the reasons for the relatively low rate of success is the impact of unmeasured technological or biological confounders. These artifacts are collectively referred to as “batch effects” because the processing date, or batch, is the most commonly measured surrogate for these unmeasured variables in genomic studies (Scharpf et al., 2011; Johnson, Li & Rabinovic, 2007; Walker et al., 2008). The umbrella term batch effects also refers to any unmeasured variables that can vary from experiment to experiment, ranging from the technician who performs the experiment to the temperature and ozone levels that day (Lander, 1999; Fare et al., 2003).

Batch effects are responsible for the failure of promising genomic prognostic signatures (Baggerly, Morris & Coombes, 2004; Baggerly, Coombes & Morris, 2005), major ambiguities in published genomic results (Spielman et al., 2007; Akey et al., 2007), and retractions of widely-publicized findings (Sebastiani et al., 2011; Lambert & Black, 2012). In many experiments, the signal from these unmeasured confounders is larger than the biological signal of interest (Leek et al., 2010). But the impact of batch effects on prediction problems has only recently been demonstrated (Parker & Leek, 2012; Luo et al., 2010). Batch effects were also recognized as a significant hurdle in the development of personalized genomic biomarkers in the Institute of Medicine’s report on clinical genomics (Micheel, Nass & Omenn, 2012).

While a number of methods have been developed for removing batch effects in population-based genomic studies (Johnson, Li & Rabinovic, 2007; Gagnon-Bartsch & Speed, 2012; Leek & Storey, 2007; Leek et al., 2010; Walker et al., 2008), there is currently no method for removing batch effects for prediction problems. There are two key differences between population level corrections and corrections designed for prediction problems. First, population level corrections assume that the biological groups of interest are known in advance. In prediction problems, the goal is to predict the biological group. Second, in prediction problems, new samples are observed one at a time, so the surrogate batch variable will have a unique and unknown value for each sample.

Here we propose frozen Surrogate Variable Analysis (fSVA) as a method for batch correction in prediction problems. fSVA borrows strength from a reference database to address the challenges unique to batch correction for prediction. The fSVA approach has two main components. First, surrogate variable analysis (SVA) is used to correct for batch effects in the training database. Any standard classification algorithm can then be applied to build a classifier based on this clean training data set. Second, probability weights and coefficients estimated on the training database are used to remove batch effects in new samples. The classifier trained on the clean database can then be applied to these cleaned samples for prediction.

We show with simulated data that the fSVA approach leads to substantial improvement in predictive accuracy when unmeasured variables are correlated with biological outcomes. We also apply fSVA to multiple publicly available microarray data sets and show improvements in prediction accuracy after correcting for batch. The methods developed in this paper have been implemented in the freely available sva Bioconductor package.

Frozen Surrogate Variable Analysis Methodology

Removing batch effects from the training set

The first step in batch correction for prediction problems is to remove batch effects from the training set. In the training set, the biological groups are known. This setting is similar to the population genomics setting and we can use a model for gene expression data originally developed for population correction of unmeasured confounders. If there are m measured features and n samples in the training set, we let Xm×n be the matrix of feature data, where xij is the value of feature i for sample j. For convenience, we will refer to X as the expression matrix for the remainder of the paper. However, our methods can be generally applied to any set of features, including measures of protein abundance, gene expression, or DNA methylation.

We propose a linear model for the relationship between the expression levels and the outcome of interest yj: xij=b0+∑k=1p1skyj+eij. The sk(⋅) are a set of basis functions parameterizing the relationship between expression and outcome. If the prediction problem is two-class, then p1 = 1 and s1(yj) = 1(yj = 1) is an indicator function that sample j belongs to class one. In a multi-class prediction problem k > 1 and the sk(⋅) may represent a factor model for each class. In matrix form this model can be written (Leek & Storey, 2007; Leek & Storey, 2008). (1) X=BS+E

where Sp1×n is a model matrix of p1 biological variables of interest for the n samples, Bm×p1 are the coefficients for these variables, and Em×n is the matrix of errors.

In genomic studies, the error term E is not independent across samples (Johnson, Li & Rabinovic, 2007; Leek & Storey, 2007; Leek & Storey, 2008; Walker et al., 2008; Friguet, Kloareg & Causeur, 2009; Leek et al., 2010; Gagnon-Bartsch & Speed, 2012). That is, there is still correlation between rows of E after accounting for the model S. The correlation is due to unmeasured and unwanted factors such as batch. We can modify model (1) to account for these measured biological factors and unmeasured biological and non-biological factors: (2) X=BS+ΓG+U

where Gp2×n is a p2 × n random matrix, called a dependence kernel (Leek & Storey, 2008) that parameterizes the effect of unmeasured confounders, Γm×p2 is the m × p2 matrix of coefficients for G, and Um×n is the m × n matrix of independent measurement errors. We previously demonstrated that such a decomposition of the variance exists under general conditions typically satisfied in population genomic experiments (Leek & Storey, 2008).

In the training set, the biological classes are known, so S is known and fixed. But the matrices B, Γ and G must be estimated. fSVA first performs surrogate variable analysis (SVA) on the training database in order to identify surrogates for batch effects in the training samples. The training set can be “cleaned” of batch effects by regressing the effect of the surrogate variables out of the data for each feature. Any classification algorithm can then be developed on the basis of the clean training data set.

SVA is an iterative algorithm that alternates between two steps. First SVA estimates the probabilities πiγ=Prγi⋅≠0→|X,S,Gˆ,πib=Prbi⋅≠0→|γi⋅≠0,X,S,Gˆ using an empirical Bayes’ estimation procedure (Leek & Storey, 2008; Efron, 2004; Storey, Akey & Kruglyak, 2005). These probabilities are then combined to define an estimate of the probability that a gene is associated with unmeasured confounders, but not with the group outcome πiw=Prbi⋅=0→&γi⋅≠0→|X,S,Gˆ=Prbi⋅=0→|γi⋅≠0,X,S,GˆPrγi⋅≠0→|X,S,Gˆ=1−πibπiγ.

The second step of the SVA algorithm weighs each row of the expression matrix X by the corresponding probability weight πˆiw and performs a singular value decomposition of the weighted matrix. Letting wˆii=πˆiw the decomposition can be written WˆX=UDVT. After iterating between these two steps, the first p2 weighted left singular vectors of X are used as estimates of G. An estimate of p2 can be obtained either through permutation (Buja & Eyuboglu, 1992) or asymptotic (Leek, 2011) approaches.

Once estimates Gˆ have been obtained, it is possible to fit the regression model in Eq. (2) using standard least squares. The result are estimates for the coefficients Bˆ and Γˆ. Batch effects can be removed from the training set by setting Xˆclean=X−ΓˆGˆ. Any standard prediction algorithm can then be applied to Xˆclean to develop a classifier based on batch-free genomic data. The result is a prediction function fXˆ⋅jclean that predicts the outcome variable yj based on the clean expression matrix.

Removing batch effects from new samples

Removing batch effects from the training database is accomplished using standard population genomic SVA batch correction. But the application of classifiers to new genomic samples requires batch correction of individual samples when both the batch and outcome variables are unknown. The fSVA algorithm borrows strength from the training database to perform this batch correction.

To remove batch effects from a new sample X⋅j′, it is first appended to the training data to create an augmented expression matrix Xj′ = [XX⋅j′] where [⋅] denotes concatenation of columns. To estimate the values of G for the new sample, fSVA uses a weighted singular value decomposition, using the probability weights estimated from the training database WˆXj′=Uj′Dj′Vj′T. The result is an estimate Gˆj′ that includes a column for the new sample. Note that only one new sample was appended. Had all the new samples been appended at once, the singular value decomposition would be highly influenced by the similarity in the new samples, rather than detecting similarities between the new sample and the database samples.

To remove batch effects from the new sample, fSVA uses the coefficients estimated from the training database Γˆ and the estimated surrogate variables: Xˆcleanj′=Xj′−ΓˆGˆj′. If there are n training samples, then the (n + 1)st column of Xclean j′ represents the new clean sample. The classifier built on the clean training data can be applied to this clean data set to classify the new sample.

Fast fSVA Methodology

fSVA requires that a new singular value decomposition be applied to the augmented expression matrix once for each new sample. Although this is somewhat computationally intensive, in typical personalized medicine applications, sample collection and processing will be spread over a long period of time. In this setting, computational time is not of critical concern. However, for evaluating the fSVA methodology or developing new classifiers using cross-validation, it is important to be able to quickly calculate clean expression values for test samples.

We propose an approximate fSVA algorithm that greatly reduces computing time by performing a streaming singular value decomposition (Warmuth & Kuzmin, 2007; Warmuth & Kuzmin, 2008). The basic idea behind our computation speed-up is to perform the singular value decomposition once on the training data, save the left singular vectors and singular values, and use them to calculate approximate values for the right singular values in new samples.

When removing batch effects from the training data, the last step is a weighted singular value decomposition of the training expression matrix WX = UDVT. After convergence, the first p2 columns of the matrix V are the surrogate variables for the training set. Since U and V are orthonormal matrices, we can write VT = D−1UTWX. The matrix P = D−1UTW projects the columns of X onto the right singular vectors VT. Pre-multiplying a set of new samples Xnew by P results in an estimate of the singular values for the new samples: VˆTnew=PTXnew. The surrogate variable estimates for the new samples consist of the first p2 columns of VˆTnew. We obtain clean data for the new samples using the estimated coefficients from the training set, identical to the calculation for the exact fSVA algorithm: Xˆclean,new=Xnew−ΓˆGˆnew.

Estimates obtained using this approximate algorithm are not identical to those obtained using the exact fSVA algorithm. The projection matrix used in the approximation, PT, is calculated using only the samples in the training set. However, there is only a one-sample difference between the projection calculated in the training set and the projection that would be obtained with exact fSVA. As the training set size grows, the approximation is closer and closer to the answer that would be obtained from the exact algorithm. For smaller databases, there is less computational burden in calculating the exact estimates. However, for large training sets, the computational savings can be dramatic, as described in the simulation below.

Simulation results

We performed a simulation to examine the benefit of fSVA in prediction problems. In order to do this, we simulated data using Eq. (2) under different distributions of each parameter. We also created discrete probability weights πiγ and πib, each equal to 1 to indicate batch- or outcome-affected, and 0 to indicate otherwise. We also varied the distribution of these probability weights (Table 1). We crafted these simulations to mimic scenarios with a subtle outcome and a strong batch effect, which is frequently the case in genomic data.

Table 1 Specifications for the three simulation scenarios used to show the performance of fSVA.

We performed three simulations under slightly different parameterizations to show the effectiveness of fSVA in improving prediction accuracy. Parameters from Eq. (2) were simulated using the distributions specified in this table. Additionally, the percentage of features in the simulation affected by batch, outcome, or both are as indicated in this table. Results from these simulations can be found in Fig. 1.

Parameter distributions	
Scenario 1	B ∼ N(0, 1)	
Γ ∼ N(0, 3)	
U ∼ N(0, 2)	
Scenario 2	B ∼ N(0, 1)	
Γ ∼ N(0, 4)	
U ∼ N(0, 3)	
Scenario 3	B ∼ N(0, 1)	
Γ ∼ N(0, 4)	
U ∼ N(0, 3)	
Affected features	
Scenario 1	50% batch-affected	
50% outcome-affected	
40% affected by both	
Scenario 2	50% batch-affected	
50% outcome-affected	
40% affected by both	
Scenario 3	80% batch-affected	
80% outcome-affected	
50% affected by both	

Figure 1 fSVA improves prediction accuracy of simulated datasets.

We created simulated datasets (consisting of a database and new samples) using model (2) and tested the prediction accuracy of these using R. For each simulated data set we performed either exact fSVA correction, fast fSVA correction, SVA correction on the database only, or no correction. We performed 100 iterations on each simulation scenario described in Table 1. We performed the simulation for a range of potential values for the correlation between the outcome we were predicting and the batch effects (x-axis in each plot). These plots show the 100 iterations, as well as the average trend line for each of the four methods investigated.

We also specified that both the simulated database and the simulated new samples have two batches and two outcomes. Each outcome was represented in 50% of the samples in both the database and the new samples. Similarly, each batch was represented in 50% of the database and new samples.

In the database, we varied the amount of confounding between batch and outcome from a Pearson’s correlation of 0 to a correlation of over 0.90. This mimics common database structures in publicly available repositories. Since the new samples are simulating a collection of single samples (such as new patients coming to the doctor), the correlation of batch and outcome within the new samples matrix is unimportant. To have a representative amount of new samples from each combination of batch and outcome, we found it best to simulate the new samples by leaving the batch and outcome uncorrelated. That way, each of the four test-cases of batch and outcome combinations was represented in 25% of the new samples.

We simulated 100 database samples and 100 new samples using the parameters described above. Each sample had 10,000 features. As a control, for each iteration in addition to performing fSVA correction, we performed SVA correction on the simulated database alone, and also performed prediction with no batch correction on the simulated database or new samples.

To quantify the effect that fSVA had on prediction, we performed exact fSVA as described above on the simulated database and new samples. We then performed Prediction Analysis of Microarrays (PAM), a commonly used method for classifying microarrays (Tibshirani et al., 2002). The PAM prediction model was built on the SVA-corrected database, and then used to predict the outcomes on the fSVA-corrected new samples. Each simulation was repeated 100 times for robustness. We performed the simulation for a range of potential values for the correlation between the outcome we were predicting and the batch effects. The correlation quantifies how much the outcome and the batch effects overlap in the training set. When the correlation is zero, the batch effects and outcome are perfectly orthogonal. When the correlation is one, then the batch effect and outcome are the same in the training set.

We found that in general the prediction accuracy measures for different iterations of a simulation varied highly, but the ordinality remained relatively constant. Therefore to display results we randomly selected three graphs from each of the scenarios, using the sample function in R (Fig. 1). Each of the graphs from the 100 iterations for each scenario can be found on the author’s website.

We found that fSVA improved the prediction accuracy in all of our simulations (Fig. 1). Interestingly, exact fSVA generally outperformed fast fSVA at all of the correlation levels except the highest correlation levels. However both fSVA methods out-performed our control of performing SVA on the database alone. Additionally any method of batch-correction generally outperformed no batch correction whatsoever.

When the batch and outcome were not correlated with each other, we saw ambiguous performance from using fSVA. This is not unexpected since it has been shown that in scenarios with no confounding between batch and outcome, batch has a minimal effect on prediction accuracy (Parker & Leek, 2012). When databases had extreme confounding between batch and outcome (above 0.85) we saw the benefits of all the batch-correction methods drop off. This is because in these situations, SVA on the database cannot differentiate batch and outcome in the database.

While in each of the simulations there was an accuracy cost to using fast fSVA vs. exact fSVA, the computational time savings was dramatic. In the scenario described, with 100 samples in the database and 100 new samples, the wall-clock computational time using a standard desktop computer for exact fSVA was 133.9 s, vs. just 1.3 s for fast fSVA. Using 50 samples in the database and 50 new samples, exact fSVA required 17.9 s vs. 0.4 s for fast fSVA. We encourage users to consider both the accuracy and computational times when selecting which algorithm to use for a particular data set.

Results from microarray studies

We examined the effect that fSVA had on several microarray studies, obtained from the Gene Expression Omnibus (GEO) website (Edgar, Domrachev & Lash, 2002). All except three of the studies were preprocessed/standardized as described previously. Three of the studies (GSE2034, GSE2603, GSE2990) were obtained from GEO and fRMA-normalized.

Each of the studies was randomly divided into equally-sized “database” and “new sample” subsets. We SVA-corrected the database subset, and then built a predictive model (PAM) on that corrected data. We then performed fSVA correction on the new samples. After performing fSVA correction, we measured the prediction accuracy of the model built on the database by calculating the number of times that the predicted outcome equaled the true outcome status, divided by the number of samples. This process was iterated 100 times for each study to obtain confidence intervals. This method is virtually identical to the simulation described above.

Results from this process can be found below (Table 2). Five of the studies showed significant improvement using fSVA. One study showed marginal improvement, with its 95% confidence interval overlapped zero. Three studies showed a cost to using fSVA, though in all three cases the 95% confidence interval for the true cost overlapped zero.

Table 2 fSVA improves prediction accuracy in 5 of the 9 studies examined.

The remaining 4 studies showed indeterminate results since the 95% confidence intervals overlapped zero. In order to find the prediction accuracy results, each of the studies was randomly divided into “database samples” and “new samples”. Exact fSVA-correction was then performed as described above. We then built a predictive model (PAM) on the database and tested the prediction accuracy on the new samples.

Study	No correction	Improvement
with fSVA	
GSE10927	0.97 (0.96, 0.97)	0.02 (0.01, 0.02)	
GSE13041	0.61 (0.59, 0.63)	0.07 (0.05, 0.10)	
GSE13911	0.93 (0.93, 0.94)	0.01 (0.00, 0.01)	
GSE2034	0.51 (0.49, 0.52)	0.03 (0.01, 0.05)	
GSE2603	0.68 (0.66, 0.70)	−0.02 (−0.04, 0.00)	
GSE2990	0.59 (0.58, 0.61)	−0.02 (−0.04, 0.00)	
GSE4183	0.89 (0.88, 0.91)	−0.02 (−0.03, 0.00)	
GSE6764	0.74 (0.72, 0.76)	0.01 (−0.01, 0.03)	
GSE7696	0.78 (0.76, 0.79)	0.02 (0.01, 0.04)	

Conclusions

Batch effects have been recognized as a crucial hurdle for population genomics experiments (Leek et al., 2010; Parker & Leek, 2012). They have also been recognized as a critical hurdle in developing genomic signatures for personalized medicine (Micheel, Nass & Omenn, 2012). Here we have introduced the first batch correction method specifically developed for prediction problems. Our approach borrows strength from a training set to infer and remove batch effects in individual clinical samples.

We have demonstrated the power of our approach in both simulated and real gene expression microarray data. However, our approach depends on similarity between the training set and the test samples, both in terms of the genes affected and the estimated coefficients. In small training sets, these assumptions may be violated. Similarly, training sets that show near perfect correlation between batch variables and biological classes represent an extreme case that can not be directly corrected using fSVA. An interesting avenue for future research is the use of publicly available microarray data to build increasingly large training databases for batch removal. We note that when the correlation between batch effects and the outcome is high it is impossible to remove these effects entirely and this may lead to poor prediction performance (Chikina & Sealfon, 2014). Therefore, it is critical to create properly designed training sets where batch effects and the outcome are not highly correlated with each other.

We have discussed here applications of fSVA to microarray data but the methodology may also be useful for other applications where genomic technologies are being applied for prediction or classification. For example, with appropriate transformations, SVA can be applied to RNA-sequencing (Leek, 2014) or DNA methylation (Jaffe et al., 2012) data. The methods we have developed here are available as part of the sva Bioconductor package (Leek et al., 2012). Code and data to reproduce this project are available at https://github.com/hilaryparker/fSVA.

Additional Information and Declarations

Competing Interests

Author Contributions

Data Deposition

The authors declare there are no competing interests.

Hilary S. Parker conceived and designed the experiments, performed the experiments, analyzed the data, contributed reagents/materials/analysis tools, wrote the paper, prepared figures and/or tables, reviewed drafts of the paper.

Héctor Corrada Bravo contributed reagents/materials/analysis tools, wrote the paper, reviewed drafts of the paper.

Jeffrey T. Leek conceived and designed the experiments, contributed reagents/materials/analysis tools, wrote the paper, reviewed drafts of the paper.

The following information was supplied regarding the deposition of related data:

Code to reproduce our analysis: https://github.com/hilaryparker/fSVA.

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
