# Peer review of "Removing batch effects for prediction problems with frozen surrogate variable analysis"

_PeerJ, doi:10.7717/peerj.561_

## Round 0.1 · original submission · Minor Revisions

Both reviewers were very positive about your work. However, both suggested a few ways in which you could improve your manuscript.

·

Basic reporting

No Comments

Experimental design

No Comments

Validity of the findings

No Comments

Additional comments

In “Removing batch effects for prediction problems with frozen surrogate variable analysis”, Parker, Bravo and Leek present a statistical method for correct for batch effects in a single sample. To correct for batch effects, one must estimate them, and doing so requires multiple samples; the authors suggest a path forward by applying surrogate variable analysis (SVA) to a set of training samples. Their approach, frozen SVA (fSVA), leverages estimates obtained from the training set to remove batch effects in a separate individual sample. The idea of fSVA is intuitive: similar samples may have experienced similar batch effects and so may benefit from similar corrections. This intuition is what connects SVA of the training sample to batch correcting the sample of interest, and fSVA is designed to exploit it.

I am in the unusual and awkward position of having no comments of substance. This paper is joy to read, and the method it describes is sensible and rather straightforward despite the technical details. The applications to simulated and real data are informative and demonstrate the utility of fSVA. The availability of fSVA in R suggests that its utility will be realized. If I am forced to say something critical, I suppose there might be room for discussing applications beyond microarrays. Perhaps elaborating in this way would recruit additional users, but I trust the authors to calibrate their narrative as they see fit.

·

Basic reporting

Minor comment:
I would suggest explaining what the correlation on the x-axis of figure 1 is in the figure legend. It is clear from the text but it would be helpful if the figure was self consistent.

Experimental design

No comments

Validity of the findings

No comments

Additional comments

Being able to use SVA for prediction is an important problem and one that deserves more attention. This paper is a valuable contribution and the approximate fSVA algorithm is a welcome addition that should aid in the adoption of this methodology.
While this may be outside the scope of this paper I would consider adding a discussion of why fSVA might lower prediction accuracy when evaluated by cross validation, for example in cases when batch effects are highly correlated with the outcome of interest. This issue is very likely to arise and it is important for the audience to realize that in such cases prediction accuracy may still be improved when evaluating "future" samples.

---

## Round 0.2 · accepted · Accept

Thank you for carefully addressing all the reviewer comments, even if minor.